# ALIGNING VISION-LANGUAGE MODELS WITH HUMAN DIRECTIONAL REFERENCES

## ABSTRACT

Spatial expressions are inherently ambiguous because communicators may adopt different perspectives, making interpretation highly dependent on the chosen frame of reference. Despite recent advances, current vision-language models (VLMs) still struggle to resolve this ambiguity in the absence of a clear reference frame, limiting effective communication between humans and machines. In contrast, humans often overcome this challenge by employing object-centered frames anchored to objects with an intrinsic *'front'*, a property known as frontedness, which determines their orientation and the spatial relationships around them. In this paper, we investigate the feasibility of endowing VLMs with object-centered spatial reasoning abilities, with frontedness as an essential component of the object-centric frame. To this end, we introduce a benchmark of synthetic 3D scenes for systematically evaluating the spatial reasoning of VLMs, and find that they consistently misidentify object orientations and tend to adopt a view-centric perspective. We show that enabling VLMs to perform spatial reasoning from an object-centric perspective achieves better alignment with human behavior.

## 1 INTRODUCTION

Recent vision-language models (VLMs) serve as a foundational backbone for general-purpose physical AI agents, positioning spatial language understanding an an essential component for human-machine interactions in the real world (Chen et al., 2024; Cheng et al., 2024). However, spatial expressions are inherently ambiguous, as their interpretation depends on the chosen *frame of reference* (FoR)–the coordinate system that enables humans to linguistically encode spatial relationships between objects (Levinson, 1996; Frank, 1998). When interpreting the scene in Figure 1, the same spatial configuration can yield different descriptions depending on the viewpoint, illustrating a viewer-centered (VC) frame anchored to the observer's perspective. In human communicative systems, this ambiguity is resolved through the use of an object-centered (OC) frame that anchors spatial descriptions to the intrinsic orientation of a reference object (Johannsen & de Ruiter, 2013; Dobnik et al., 2014). Adopting an OC frame allows humans to express consistent spatial descriptions across different viewpoints (Mou & McNamara, 2002; Levinson, 2003; Li et al., 2011).

Central importance to the OC frame adoption is the presence of *frontedness*, which defines the object's inherent orientation (Carlson-Radvansky & Logan, 1997; Levinson, 2003; Robinette et al., 2010). Frontedness is determined by two key dimensions: a visual property reflecting asymmetric appearances and a functional property arising from human-object interactions (Eschenbach, 2004; Harris, 2024). For example, people intuitively treat the side of a sofa with seating space as its front, based on functional expectations combined with visual cues. The salient frontedness encourages greater use of the OC frame (Robinette et al., 2010). While recent works have attempted to extend spatial reasoning from limited viewpoints (Lee et al., 2025; Daxberger et al., 2025; Yin et al., 2025; Xu et al., 2025), it remains unexplored whether current VLMs align with human strategies for resolving FoR-related ambiguity based on frontedness.

To bridge the gap, we introduce **S**patial relation and **O**rientation with **F**rontedness **A**ssessment (SOFA), a benchmark that evaluates the spatial reasoning capabilities of VLMs with a focus on object-centered perspective. We construct synthetic 3D scenes featuring realistic objects in controlled spatial configurations, paired with spatial reasoning queries. SOFA consists of two complementary tasks: (1) object orientation understanding, which investigates whether models can identify

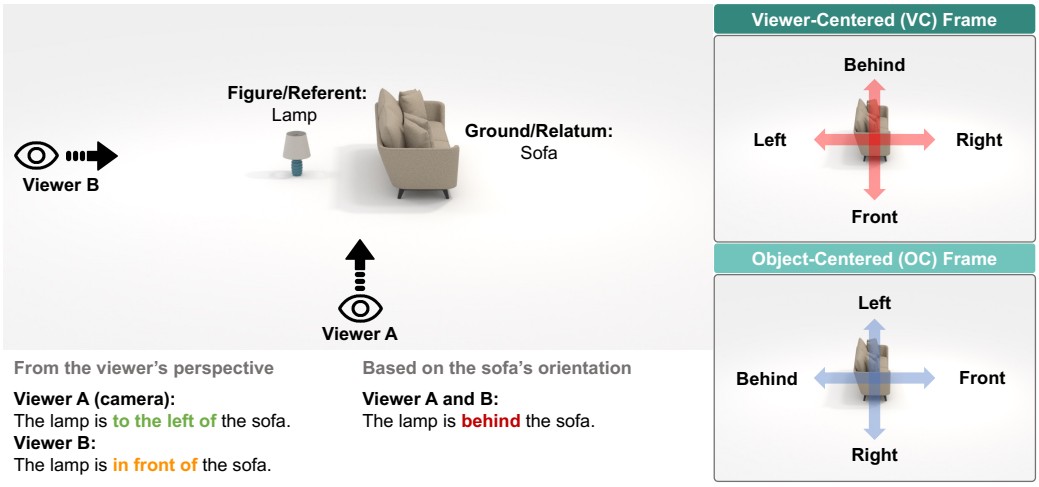

Figure 1: Frame of reference selection. The choice of reference frame determines spatial descriptions: OC frames provide consistent relations across viewpoints while VC frames vary with the observer's perspective. The examples show how spatial relations are defined by VC frame from camera perspective (top) versus OC frame anchored to the sofa's intrinsic orientation (bottom).

object orientation in the presence of an intrinsic front, and (2) spatial relation reasoning, which assesses whether models can select the correct direction with appropriate FoR based on object properties. This framework allows systematic evaluation of how VLMs resolve FoR-related ambiguity—through interpreting frontedness and adopting OC frames—under controlled synthetic scenes and structured linguistic queries (see Figure 2).

Our evaluation of VLMs' spatial reasoning capabilities using SOFA reveals key findings. First, current VLMs often fail to reliably identify object orientations, indicating an inadequate understanding of frontedness. Second, even when orientation cues are visually salient, the models overwhelmingly default to viewer-centric descriptions, reaffirming a persistent bias toward the camera's viewpoint. These results underscore a deeper lack of flexibility in frame selection and highlight the absence of inductive mechanisms necessary for adopting OC frames in contextually appropriate situations.

Building on these findings, we investigate whether models can be endowed with the ability to perform spatial reasoning in the OC framework by leveraging the concept of frontedness. Fine-tuned on training data annotated in the same way as our SOFA, the VLMs approach human performance in orientation and spatial relation understanding and show better alignment with human behavior.

In summary, our contributions are as follows:

1. We introduce SOFA, a benchmark for evaluating the spatial reasoning capabilities of VLMs in the object-centric perspective.

2. Evaluation of VLMs on SOFA reveals limitations in their understanding of orientation and a tendency to adopt the viewer-centric perspective in spatial relation reasoning.

3. Our experiments demonstrate that fine-tuned VLMs can adaptively perform spatial reasoning in the object-centric perspective, showing closer alignment with human spatial reasoning.

## 2 BACKGROUND AND RELATED WORK

### 2.1 OBJECT-CENTERED FRAME OF REFERENCE

Frame of reference (FoR) is a coordinate system for describing spatial relations, providing a linguistic structure that encodes the position of a figure (referent) relative to a ground (relatum) (Levinson, 1996; Frank, 1998). Levinson (2003) has classified FoR into three major types: absolute, relative, and intrinsic. The absolute FoR provides a geocentric framework based on cardinal directions, such as north and south. The relative FoR, also termed viewer-centered (VC) frame, describes the position of the figure relative to the ground from the observer's perspective. The intrinsic FoR, or object-

centered (OC) frame, defines the position of the figure with the ground's intrinsic orientation as the origin. Depending on the selection of FoRs, the same configuration can yield divergent descriptions, which makes spatial expressions inherently ambiguous. Although humans frequently adopt VC frames in situated communication, their reliance on perspective introduces inconsistency (Schober, 1996). A particularly robust strategy to overcome this limitation is the use of OC frames, which provide viewpoint-invariant spatial descriptions (Ziegler et al., 2012; Johannsen & de Ruiter, 2013; Dobnik et al., 2014).

A key prerequisite for adopting an OC frame is the ability to identify frontedness—the presence of an intrinsic 'front' that defines the orientation of an object (Levinson, 1996; Barbara & Jackendoff, 1993; Robinette et al., 2010). Frontedness arises from two complementary properties: (1) *visual* property, which reflects geometric asymmetry, and (2) *functional* property, which emerges from how humans interact with objects in everyday contexts (Eschenbach, 2004; Harris, 2024). When the ground object exhibits salient frontedness, it can serve as a stable anchor for spatial descriptions. The adoption of OC frames provides two key advantages. Firstly, in conversation, it reduces ambiguity by anchoring spatial descriptions to the intrinsic orientation of objects, which helps interlocutors achieve consistent interpretations (Dobnik et al., 2014). Secondly, in spatial memory, OC frames enable more accurate recall of the spatial layout within the surrounding environment, constructing relations aligned with the object's intrinsic axes (Mou & McNamara, 2002; Li et al., 2011).

### 2.2 Spatial Reasoning and Orientation Understanding in VLMs

Recent advances in vision-language models (VLMs) have exhibited the integration of linguistic and visual representations (Bai et al., 2023; OpenAI, 2024; Reid et al., 2024), highlighting spatial understanding as a core cognitive capability (Kamath et al., 2023; Liu et al., 2023; Shiri et al., 2024; Zhang et al., 2025a). Although several benchmarks have evaluated spatial understanding in VLMs from various perspectives (Ma et al., 2025; Liu et al., 2025; Zhang et al., 2025a; Yin et al., 2025), limited consideration of FoR undermines the consistency and reliability of such evaluations. Based on a systematic incorporation of FoR with synthetic 3D scenes, COMFORT (Zhang et al., 2025b) addresses the ambiguity of spatial terms and reveals that most VLMs predominantly rely on the VC frame, especially from the camera's perspective. While previous works have extended spatial reasoning beyond the camera's viewpoint and proposed approaches to simulate perspective shifts from multiple viewpoints, such as multi-view image augmentation or numerical coordinate encoding (Ray et al., 2024; Yin et al., 2025; Daxberger et al., 2025; Xu et al., 2025; Lee et al., 2025), these approaches remain misaligned with human strategies for resolving FoR-related ambiguity, which rely on an object's intrinsic orientation rather than inconsistent perspectives.

Object orientation understanding has been a fundamental task in computer vision, with pose estimation serving as a basis for interpreting object orientation (Ozuysal et al., 2009; Di et al., 2022). With the advent of VLMs, recent studies have introduced evaluation benchmarks with natural language instructions instead of pose information (Tong et al., 2024; Wang et al., 2025; Ma et al., 2025). EgoOrientBench (Jung et al., 2025) assesses object orientation comprehension by incorporating synthetic 3D images with discrete orientation classes, while DORI (Nichols et al., 2025) provides a fine-grained framework with continuous angular estimations. However, previous works overlook the intrinsic property of objects—frontedness—that defines their orientation. To bridge this gap, we systematically examine models' cognitive ability to understand object orientation and adopt appropriate FoR based on frontedness.

## 3 SOFA: Spatial Relation and Orientation with Frontedness Assessment

We introduce the **S**patial relation and **O**rientation with **F**rontedness **A**ssessment (SOFA), a benchmark designed to assess VLMs' ability to linguistically reconstruct spatial configurations from the OC perspective. SOFA consists of object orientation understanding and spatial relation reasoning tasks. Each task employs visual question answering format with image-text pairs, requiring models to translate spatial visual information into coherent linguistic expressions. For systematic measurement of models' competence, we carefully design synthetic visual scenes with controlled spatial configurations and corresponding textual prompts. Based on this design principle, SOFA estab-

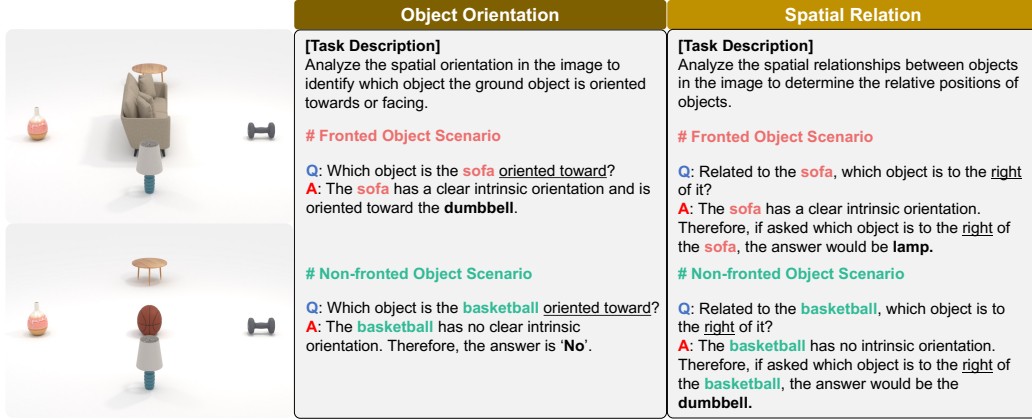

Figure 2: SOFA Benchmark. In the fronted object (e.g., sofa) scenarios, VLMs identify the ground's orientation and adopts OC frame to describe spatial relationships. In the non-fronted object (e.g., basketball) scenarios, they recognize the ground's lack of frontedness and use VC frame to describe spatial relationships.

lishes a unified evaluation framework for VLMs' proficiency in the FoR adoption, bridging object properties and spatial comprehension (Figure 2).

## 3.1 SCENE CONSTRUCTION

For visual contexts, we construct synthetic 3D scenes with object meshes from ABO (Collins et al., 2022) and COMFORT (Zhang et al., 2025b), rendering them into images in Blender (Blender Online Community, 2016). Each scene illustrates a spatial configuration of ground and figure objects, which we define as a quadruple $(O, p, \theta, L)$. The object set $O = \{g, f_1, f_2, f_3, f_4\}$ consists of the ground $g$ and figures $f_i$. The object property $p \in \{fronted, non\text{-}fronted\}$ indicates whether $g$ possesses frontedness; when $p = fronted$, we discretize the rotation as $\theta \in \{0°, 90°, 180°, 270°\}$ so that the intrinsic front coincides with standard coordinate axes following camera convention (i.e., front, left, behind, and right). For non-fronted objects, rotation is not specified since these objects lack intrinsic orientation. When $g$ is located at $(x_g, y_g)$, the spatial layout $L$ arranges $f_i$ at four locations: $\{(x_g - d, y_g), (x_g + d, y_g), (x_g, y_g - d), (x_g, y_g + d)\}$, where $d$ denotes the distance from $g$. We select object categories from everyday contexts where frontedness is apparent through human-object interactions. Accordingly, the visual context comprises the following configuration of ground and figure objects:

- **Ground objects:** We prioritize the presence of clear frontedness, selecting 4 fronted objects (sofa, chair, bench, laptop) and 4 non-fronted objects (ottoman, waste bin, basketball, stool). Each scene includes a single ground object placed at the center. When the ground object possesses an intrinsic front, it is rotated in successive 90° increments relative to the camera-facing direction.

- **Figure objects:** The spatial layout consists of four non-fronted indoor objects (vase, lamp, dumbbell, table). To resolve ambiguity in spatial relation terms, the figure objects are placed along the camera's coordinate axes: sagittal (front/behind) and lateral (left/right). For reliable assessment, we permute the figure objects across the four canonical positions, preventing potential biases toward specific objects or locations.

## 3.2 TASK DESIGN

Based on the notion of frontedness, we formulate a twofold evaluation with complementary tasks: object orientation understanding and spatial relation reasoning. Each task evaluates a VLM through a natural language interface, grounded in the visual and linguistic contexts. The linguistic context consists of a task description and a corresponding query. Task descriptions indicate the evaluation type, thereby directing the model toward object orientation assessment or spatial relationship inference. In total, we construct our benchmark with 2,496 image-text pairs across both task types (see Appendix A for detailed examples and prompt templates).

Table 1: Quantitative comparison on the SOFA. Ori-F/NF/Adapt evaluate orientation understanding, while FoR-OC/VC/Adapt evaluate spatial relations using different reference frames.

| Model | Object Orientation | | | Spatial Relation | | |
|---|---|---|---|---|---|---|
| | Ori-F | Ori-NF | Ori-Adapt | FoR-OC | FoR-VC | FoR-Adapt |
| Random | 0.200 | 0.200 | 0.200 | 0.250 | 0.250 | 0.250 |
| Human | 1.000 | 1.000 | 1.000 | 0.893 | 1.000 | 0.920 |
| *Open-sourced models* | | | | | | |
| InstructBLIP-7B | 0.206 | 0.011 | 0.167 | 0.223 | 0.227 | 0.224 |
| InstructBLIP-13B | 0.152 | 0.026 | 0.127 | 0.155 | 0.138 | 0.151 |
| LLaVA-NeXT-8B | 0.254 | 0.073 | 0.219 | 0.099 | 0.398 | 0.174 |
| LLaVA-NeXT-72B | 0.264 | 0.250 | 0.261 | 0.082 | 0.453 | 0.174 |
| InternVL3-8B | 0.256 | 0.099 | 0.222 | 0.103 | 0.809 | 0.279 |
| InternVL3-78B | 0.333 | 0.474 | 0.361 | 0.005 | 0.968 | 0.246 |
| Qwen2.5-VL-7B | 0.197 | 0.604 | 0.278 | 0.053 | 0.776 | 0.234 |
| Qwen2.5-VL-72B | 0.315 | 0.911 | 0.434 | 0.014 | 0.825 | 0.217 |
| *Proprietary models* | | | | | | |
| GPT-4o | 0.337 | 0.760 | 0.422 | 0.047 | 0.828 | 0.242 |
| Gemini-2.0-Flash | 0.242 | 0.146 | 0.223 | 0.006 | 0.924 | 0.236 |
| **Ours (LLaVA-NeXT-8B)** | 0.695 | 0.797 | 0.716 | 0.789 | 0.747 | 0.778 |
| **Ours (Qwen2.5-VL-7B)** | 0.839 | 0.989 | 0.869 | 0.727 | 0.904 | 0.771 |

**Object Orientation Understanding.** Object orientation queries ask which figure object the ground object is oriented toward. Depending on the ground object's property, models should identify the specific figure object located in the facing direction for fronted objects, or respond "No" for non-fronted objects that lack intrinsic orientation.

**Spatial Relation Reasoning.** Spatial relation queries specify which figure object satisfies a spatial relationship with the ground object, leaving the FoR unstated to test whether the model adopts the appropriate reference frame based on frontedness. The visual context configures the ground object's orientation axes to correspond with the camera's coordinate systems; therefore, we focus on four prototypical spatial terms–*front*, *behind*, *left*, and *right*–with figure objects positioned along these axes.

### 3.3 Evaluation Metric

We use accuracy as the main evaluation metric because the labels are evenly distributed. Given the visual context $x_i^{\text{img}}$ and the linguistic context $x_i^{\text{text}}$, the VLM $\mathcal{M}$ predicts one of the figure objects as its response; formally expressed as $\mathcal{M}(x_i^{\text{img}}, x_i^{\text{text}}) = y_i$, where $y_i \in \{f_1, f_2, f_3, f_4\}$. Instead of binary responses or spatial terms, the model prediction is an object category and we adopt this response format for two reasons: (1) to prevent potential bias toward affirmative responses (e.g., "yes"; Dentella et al., 2023) and (2) to avoid preference for predicting specific spatial terms (e.g., "left"; Chen et al., 2025) regardless of visual contexts.

**Object Orientation Understanding.** We include non-fronted ground objects, for which the model must recognize the absence of orientation (represented by $\emptyset$). Accordingly, the metric for orientation understanding is defined as:

$$\text{Ori-Adapt} = \frac{1}{N} \sum_{i=1}^{N} \begin{cases} \mathbf{1}\{\hat{y}_i = y_i\} & \text{if } p_i = \textit{fronted}, \\ \mathbf{1}\{\hat{y}_i = \emptyset\} & \text{if } p_i = \textit{non-fronted}, \end{cases}$$

where $y_i$ denotes the ground truth object and $\hat{y}_i$ the predicted one. $N$ is the number of evaluated image–text pairs. Ori-F measures the ability to identify correct orientation for fronted objects, while Ori-NF evaluates recognition of the lack of intrinsic orientation for non-fronted objects.

**Spatial Relation Reasoning.** Since spatial relation interpretation fundamentally depends on the selected reference frame, we evaluate models' adaptability in choosing the appropriate FoR. Once

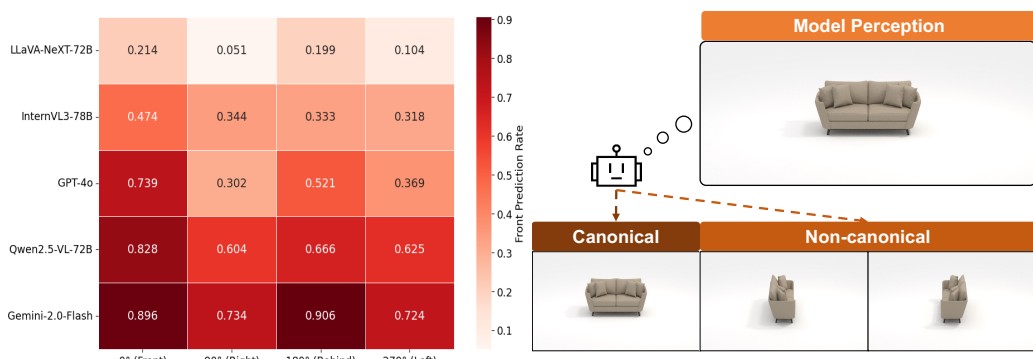

Figure 3: Canonical bias in orientation prediction. Heatmap shows most VLMs demonstrate systematic preference for frontal orientation regardless of actual object rotation. This indicates models interpret objects through default canonical views (front-facing) instead of understanding object orientations.

frontedness is recognized, models must determine whether to adopt a viewer-centric or object-centric perspective. Objects with frontedness provide intrinsic directional axes that enable object-centric spatial reasoning, leading to the OC frame as the natural choice. In contrast, non-fronted objects lack such directional cues, necessitating reliance on the VC frame. Under varying ground truth across different FoRs, we systematically measure the model's FoR adaptation capability by

$$\text{FoR-Adapt} = \frac{1}{N} \sum_{i=1}^{N} \begin{cases} \mathbf{1}\{\hat{y}_i = y_i^{\text{oc}}\} & \text{if } p_i = \textit{fronted}, \\ \mathbf{1}\{\hat{y}_i = y_i^{\text{vc}}\} & \text{if } p_i = \textit{non-fronted}, \end{cases}$$

where $y_i^{\text{oc}}$ and $y_i^{\text{vc}}$ denote the ground truth objects in the OC frame and VC frame, respectively. Note that we exclude cases where OC and VC frames produce identical spatial relations, since such cases are indistinguishable between the two frames. FoR-OC measures performance using OC frame for fronted objects, while FoR-VC evaluates performance using VC frame for non-fronted objects. Supplementary metric formulations are provided in the Appendix A.

## 4 EXPERIMENTAL RESULT AND ANALYSIS

**Baselines.** We compare a wide range of VLMs to evaluate their performance on object orientation and spatial relation understanding tasks in SOFA. Our baselines include open-source models: InstructBLIP (Dai et al., 2023), LLaVA-NeXT (Li et al., 2025), InternVL3 (Zhu et al., 2025), and Qwen2.5-VL (Bai et al., 2025). These models follow the standard VLM architecture of connecting vision encoders to pretrained language models via projection (Li et al., 2023). All baseline models are instruction-tuned models, as SOFA requires proper interpretation of natural language instructions. We also report results from proprietary models, such as GPT-4o (OpenAI, 2024) and Gemini-2.0-Flash (Reid et al., 2024). Additionally, we include human performance as an upper bound and random chance as a reference baseline for comparison. Detailed evaluation setups are provided in Appendix B.

**Training.** We fine-tune Qwen2.5-VL-7B and LLaVA-NeXT-8B for object-centric spatial reasoning, applying LoRA (Hu et al., 2022) and unfreezing the vision encoder to capture fine-grained visual features that distinguish orientational states and identify object frontedness. For the training dataset, we construct synthetic scenes and exclude any object meshes that appear in SOFA to ensure fair comparison. We conduct detailed analysis and ablation studies using Qwen2.5-VL-7B as our primary model. Training dataset and implementation details are provided in the Appendix C.

### 4.1 SPATIAL BIAS OF VLMS

Evaluations on SOFA reveal that VLMs exhibit significant biases in orientation and spatial relation understanding. These biases prevent VLMs from aligning with human spatial reference frames.

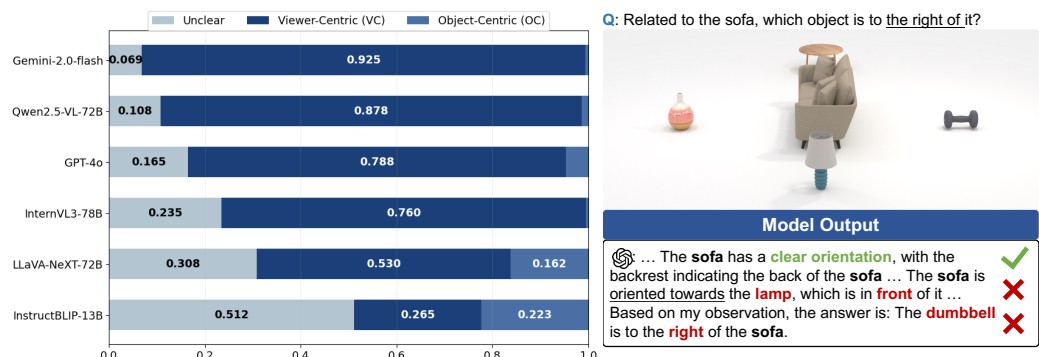

Figure 4: Viewer-centric bias in spatial reasoning. Response distributions illustrate that most VLMs rely on VC (camera) over OC perspective in scenarios where ground objects possess salient frontedness. The example shows GPT-4o recognizing the sofa's clear orientation yet defaulting to the VC frame.

**Canonical Bias.** As shown in Table 1, we observe that most VLMs exhibit very limited ability in orientation tasks, with accuracy often close to random chance. Figure 3 further shows that the models' predictions frequently focus on the front, reflecting a canonical bias reported in prior work, where objects are consistently recognized as front-facing (0°) regardless of their actual orientation (Jung et al., 2025).[1] The underlying issue is that visually distinct object rotations are often mapped to similar feature representations, making it difficult for models to distinguish between different orientational states. MMVP (Tong et al., 2024) attributes this limitation to the visual encoder's inability to adequately represent fine-grained visual details, as it prioritizes overall semantic understanding and conceptual familiarity, thereby discarding critical spatial information.

**Viewer-Centric Bias.** The results of spatial relation reasoning in Table 1 show that most VLMs struggle with spatial understanding in the OC frame, highlighting fundamental difficulties in anchoring spatial relations to ground objects. We conducted an additional experiment, which reconfirmed that models default to VC interpretations regardless of object orientation. As shown in Figure 4, most models exhibit a strong VC bias even when ground objects have clear frontedness. Consequently, due to their tendency toward the VC frame, their ability to adaptively select the FoR remains very limited. Previous works have also reported viewer-centric dependency in VLMs. They demonstrate that, while VLMs exhibit sophisticated spatial understanding within the camera coordinate system (Zhang et al., 2025b; Linsley et al., 2025; Zhang et al., 2025a), their bias toward the camera's perspective fundamentally limits their ability to perform spatial reasoning from novel viewpoints (Xu et al., 2025; Lee et al., 2025).

### 4.2 ALIGNMENT OF VLMS WITH HUMAN FoR

As shown in Table 1, our model achieves consistently high scores across all metrics, indicating that VLMs can acquire the concept of object frontedness, reason about object orientation, and perform spatial reasoning within the OC framework. Moreover, its performance comes closer to that of humans, while mitigating the biases of base VLMs.

**Adaptive Frame-of-Reference Selection.** Most notably, our models achieve the highest scores (0.771 and 0.778), approaching human-level performance and substantially outperforming all baseline models. These results suggest that VLMs possess a nuanced ability to select the appropriate frame of reference based on the intrinsic properties of ground objects, in alignment with human performance patterns on FoR-OC and FoR-VC, which measure the accuracy of selecting the correct target objects in the OC and VC frames, respectively.

**Common Confusion in Ambiguous Cases.** In Table 2, the lowest OC performance at 0° (0.276) reflects inherent lateral-axis confusion when objects are front-facing, mirroring human cognitive

---

[1]A canonical view refers to the most typical and representative viewpoint of an object—one that maximally reveals its distinctive features and corresponds to how people naturally prefer to see or depict that object (Blanz et al., 1999).

Table 2: Ablation results for training objectives. The experimental results demonstrate performance across object orientations (0°–270°) for models trained with different objectives: orientation prediction (+Orient.), object-centric spatial reasoning (+OC), both combined (Ours), and baseline (Base).

| Model | 0° | | 90° | | 180° | | 270° | | Average | |
|---|---|---|---|---|---|---|---|---|---|---|
| | Orient. | OC | Orient. | OC | Orient. | OC | Orient. | OC | Orient. | OC |
| Human | 1.000 | 0.537 | 1.000 | 0.950 | 1.000 | 0.975 | 1.000 | 0.975 | 1.000 | 0.893 |
| Base | 0.469 | 0.005 | 0.042 | 0.052 | 0.078 | 0.052 | 0.026 | 0.078 | 0.154 | 0.053 |
| + Orient. Train | 0.781 | 0.000 | 0.448 | 0.044 | 0.453 | 0.005 | 0.989 | 0.078 | 0.595 | 0.042 |
| + OC Train | 0.479 | 0.302 | 0.078 | 0.503 | 0.219 | 0.495 | 0.599 | 0.706 | 0.344 | 0.536 |
| Ours (+ Orient & OC) | 0.849 | 0.276 | 0.833 | 0.740 | 0.682 | 0.620 | 0.989 | 0.992 | 0.839 | 0.727 |

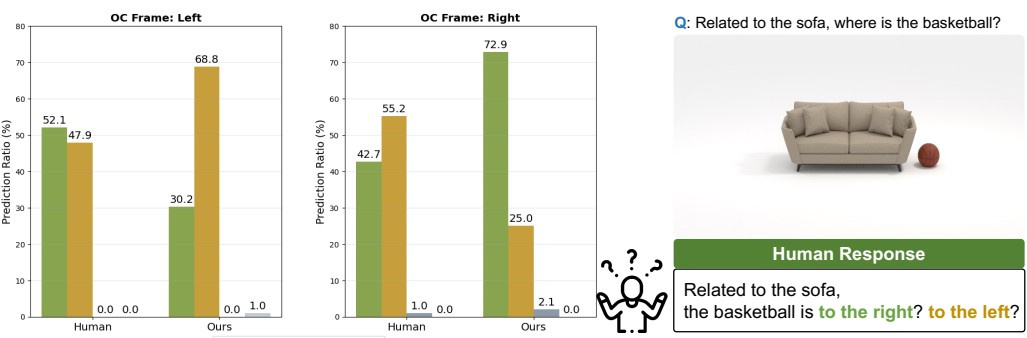

Figure 5: Shared confusion patterns in spatial reasoning. When the ground object faces forward (0°), both humans and our model struggle with lateral axis (left/right) disambiguation, demonstrating similar FoR confusion patterns.

patterns, where human performance is also lowest at this orientation. To be more specific, as shown in Figure 5, the distributions of human and model predictions show that while both avoid selecting the front–behind direction, they are often confused about the left–right direction, failing to settle on a single FoR and oscillating between OC and VC frames.

**Synergistic Effect of Orientation and OC Frame Learning.** We conduct an ablation study to examine whether the model can adopt the OC frame of reference based solely on orientation understanding. As shown in Table 2, orientation training alone does not enable the model to perceive the spatial environment in the OC frame, yielding only 0.042 on OC tasks compared to 0.595 on orientation tasks. Conversely, when trained to reason about spatial relations using OC frames, the model's performance on orientation tasks increases from 0.154 to 0.344, suggesting that comprehension of intrinsic axial structure indirectly enhances orientation understanding. Finally, training on both objectives yields the best overall performance, indicating that the two components are complementary and mutually reinforce each other during concept learning.

### 4.3 GENERALIZATION TO UNSEEN AND REAL-WORLD SCENARIOS

To examine the generalization of our approach, we evaluate the model on unseen instances from training categories (sofa, chair, ottoman, waste bin), entirely unseen categories (bench, laptop, basketball, stool), and real-world environments (i.e., real images).

**Unseen Object Instances and Types.** As shown in Table 3, spatial-relation reasoning performance on objects from unseen instances and categories shows an improvement over the base model. The model demonstrates competitive performance in both OC and VC frames, implying that the learned spatial reasoning in the adaptive FoR generalizes to unseen objects.

**Real-World Scenarios.** Since our method is trained exclusively on synthetic data, evaluation on real images provides a good indicator of generalization to real-world scenarios. We design our evaluation with the following considerations. First, as real images contain multiple objects of diverse types, we

Table 3: Category generalization on spatial reasoning tasks. Results show whether models appropriately select FoR based on different ground object categories. Pink represents seen categories (training), blue indicates unseen categories. Fronted objects evaluated with OC frame, non-fronted objects with VC frame.

| Model | Fronted Object | | | | Non-fronted Object | | | |
|---|---|---|---|---|---|---|---|---|
| | Sofa | Chair | Bench | Laptop | Ottoman | Waste bin | Basketball | Stool |
| Base | 0.052 | 0.049 | 0.056 | 0.056 | 0.781 | 0.688 | 0.865 | 0.771 |
| Ours | 0.747 | 0.826 | 0.771 | 0.562 | 0.989 | 0.885 | 0.989 | 0.750 |

Table 4: Performance comparison on real-world scenarios. Our model demonstrates robust spatial reasoning capabilities, outperforming baseline models across all four spatial relations.

| Model | Spatial Relation: OC Frame | | | | |
|---|---|---|---|---|---|
| | Front | Behind | Left | Right | Average |
| Llava-NeXT-8B | 0.086 | 0.014 | 0.243 | 0.314 | 0.164 |
| InternVL3-8B | 0.443 | 0.028 | 0.171 | 0.257 | 0.225 |
| Qwen2.5-VL-7B | 0.414 | 0.086 | 0.071 | 0.043 | 0.156 |
| Ours | **0.471** | **0.486** | **0.500** | **0.400** | **0.464** |

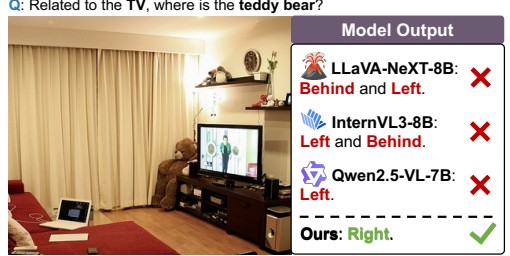

Figure 6: Example of real-world spatial reasoning performance showing model outputs for different VLMs.

formulate queries in the form *"Where is the [figure] relative to [ground]"* to specify target objects explicitly and reduce multi-object hallucination. We then evaluate spatial relationships in terms of four standard relations: *front*, *behind*, *left*, and *right*. Second, because controlled collection is challenging for real images, we restrict our evaluation to fronted objects under the OC perspective. We collect real images from MS-COCO dataset (Lin et al., 2014). For fair comparison, we balance the dataset with 70 examples per spatial relationship, totaling 280 image-text pairs. Objects include TV, refrigerator, radiator, and other items not commonly found in training data.

Table 4 illustrates that our approach demonstrates strong generalization from synthetic to real-world scenarios. Despite being trained exclusively on synthetic data with a limited set of object categories in canonical configurations, our model substantially outperforms all baseline VLMs across all spatial relationships, achieving an average accuracy of 0.464. The performance gap is especially pronounced for challenging relationships such as "behind", where other models perform poorly. LLaVA-NeXT performs better on "left" (0.243) and "right" (0.314), while struggling most with "front/behind" relationships. InternVL3 and Qwen2.5-VL achieve competent performance limited to "front" relationships, scoring 0.443 and 0.414 respectively. Detailed prediction distribution for all baseline models is provided in Appendix D.

## 5 CONCLUSION

In this work, we investigated current VLMs' inability to align with human strategies for resolving spatial ambiguity through object-centered reasoning. We introduced SOFA, a benchmark for evaluating spatial reasoning with explicit consideration of frontedness and reference frame selection. Our systematic evaluation revealed that existing models struggle with object orientation understanding, exhibit severe canonical bias, and overwhelmingly default to viewer-centric descriptions regardless of object properties. Our training methodology demonstrates that object-centric spatial reasoning can be effectively acquired through relational grounding with simplified synthetic scenarios, achieving substantial performance improvements while maintaining consistency across diverse spatial relationships. These findings highlight the importance of aligning VLMs with human spatial cognition strategies, establishing foundations for more effective human-machine spatial communication in real-world interactions.

## REPRODUCIBILITY STATEMENT

The benchmark construction process with publicly available 3D object meshes is documented in Appendix A. Evaluation details are in the Appendix B, and the training setup is in Appendix C. We include evaluation code and some example synthetic scenes in the supplementary materials. All our experiments use both open-source and proprietary models that are publicly accessible.

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

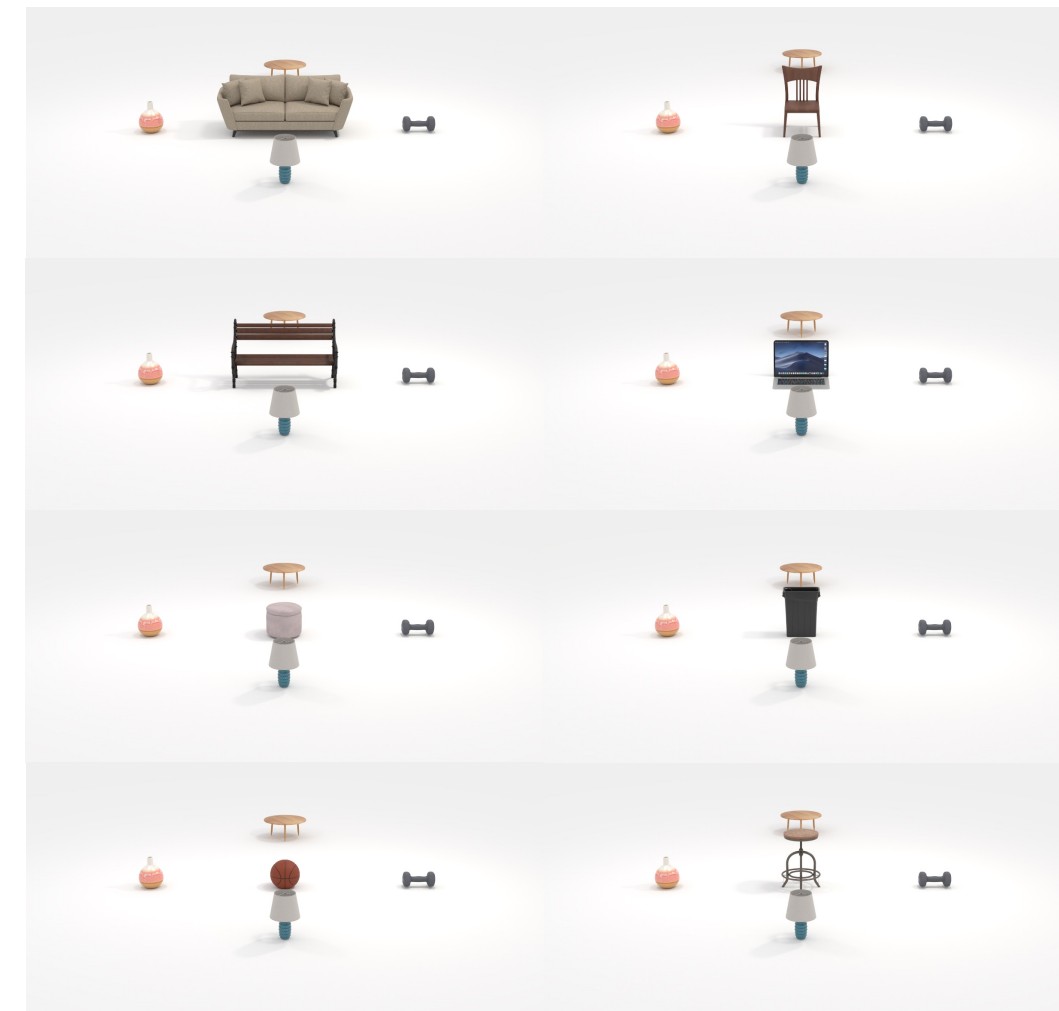

Figure 7: Representative spatial scenes for each ground object category. Top row shows fronted objects (sofa, chair, bench, laptop) with clear intrinsic orientations, bottom row displays non-fronted objects (ottoman, waste bin, basketball, stool) lacking directional cues.

## A    BENCHMARK DETAILS

### A.1    SYNTHETIC SPATIAL SCENES

Our benchmark comprises 480 synthetic images rendered in Blender (Blender Online Community, 2016), with cameras positioned at human eye-level to ensure naturalistic viewing perspectives. The dataset includes 8 ground object categories: 4 fronted objects (sofa, chair, bench, laptop) and 4 non-fronted objects (ottoman, waste bin, basketball, stool), selected from ABO (Collins et al., 2022) and COMFORT (Zhang et al., 2025b) datasets. Figure objects (vase, lamp, dumbbell, table) are positioned at four canonical locations relative to the ground object centered at $(x_g, y_g)$: front $(x_g, y_g-d)$, behind $(x_g, y_g+d)$, left $(x_g-d, y_g)$, and right $(x_g+d, y_g)$, with a fixed distance $d$ from the center. All scenes use consistent lighting conditions with ambient illumination and a single directional light source to ensure clear object visibility and consistent shadows. Representative examples of spatial scenes for each ground object category can be found in Figure 7.

### A.2    PROMPT TEMPLATES

Our benchmark consists of 2,496 image-text pairs in total, including 960 queries for object orientation understanding and 1,536 queries for spatial relation reasoning tasks. Object orientation queries use two question formats: *"Which object is the [ground object] {oriented toward / facing}?"* For

| [Task] | [Task] |
|---|---|
| Analyze the spatial orientation in the image to identify which object the ground object is oriented towards or facing. | Analyze the spatial relationships between objects in the image to determine the relative positions of objects. |
| **[Instruction]** | **[Instruction]** |
| 1. Identify the ground object and determine if it has a clear intrinsic front or orientation | 1. Identify the ground object and determine if it has a clear intrinsic front or orientation |
|   - If the ground object has no clear orientation (non-fronted), answer exactly "No" | 2. If the ground object has a specific orientation, determine which object it is positioned towards |
| 2. If the ground object has a specific orientation, determine which object it is positioned towards | 3. Focus ONLY on objects that are clearly visible in the scene |
| 3. Focus ONLY on objects that are clearly visible in the scene | 4. Analyze spatial relationship between objects the appropriate frame of reference: |
| 4. The ground object's orientation is determined by identifying which object it is directly facing or pointing towards |   - When the ground object has a clear orientation, describe spatial relation based on the ground object's orientation |
| 5. Analyze the spatial arrangement to determine which object the ground object is positioned to face |   - When the ground object has no orientation, describe spatial relation based on the camera's perspective |
| 6. Provide step-by-step reasoning before giving the final answer | 5. Provide structured spatial analysis before giving the final answer |
| **[Answer Format]** | **[Answer Format]** |
| Provide your reasoning first, then give your final answer in the following format: | Provide your reasoning first, then give your final answer in the following format: |
| Based on my observation, the answer is: <think>(Replace with your reasoning here)</think><answer>(Replace with your answer here)</answer> | Based on my observation, the answer is: <think>(Replace with your reasoning here)</think><answer>(Replace with your answer here)</answer> |
| **[Question]** | **[Question]** |
| {question} | {question} |

Figure 8: Structured prompt templates used for object orientation understanding (left) and spatial relation reasoning (right) tasks in SOFA benchmark.

spatial relation queries, we employ four directional terms: *"Related to the [ground object], which object is {in front of / behind / to the left of / to the right of} it?"* To ensure reliable evaluation of frame of reference adaptation, we exclude cases where OC and VC frames yield identical spatial relations—specifically, front/behind queries when objects are at $0°$ orientation and left/right queries when objects are at $180°$ orientation. Each synthetic image is paired with multiple task-specific queries to comprehensively evaluate VLMs' spatial reasoning capabilities based on object frontedness. The instruction templates explicitly guide models to identify object frontedness and select appropriate frames of reference for spatial reasoning accordingly. The answer format follows a structured reasoning-then-answer approach, facilitating systematic evaluation by capturing both the model's spatial reasoning process and final predictions. Detailed examples of prompt templates for both task types are provided in Figure 8.

## A.3 EVALUATION METRICS

SOFA framework quantitatively evaluates VLMs across fronted and non-fronted object scenarios, each with dedicated metrics. For fronted object scenarios, Ori-F determines whether models can interpret object orientation based on frontedness cues by requiring them to correctly identify which figure object the ground object faces. FoR-OC assesses the ability to perform spatial reasoning using OC frame by evaluating spatial relations based on the object's intrinsic orientation:

$$\text{Ori-F} = \frac{1}{N}\sum_i \mathbf{1}\{\hat{y}_i = y_i\}, \quad \text{FoR-OC} = \frac{1}{N}\sum_i \mathbf{1}\{\hat{y}_i = y_i^{\text{oc}}\} \tag{1}$$

For non-fronted object scenarios, Ori-NF measures whether models can recognize the absence of intrinsic orientation by expecting the model to respond with no orientation indication. FoR-VC evaluates spatial reasoning from VC frame by assessing spatial understanding from the camera's

viewpoint:

$$\text{Ori-NF} = \frac{1}{N} \sum_i \mathbf{1}\{\hat{y}_i = \emptyset\}, \quad \text{FoR-VC} = \frac{1}{N} \sum_i \mathbf{1}\{\hat{y}_i = y_i^{\text{vc}}\} \tag{2}$$

The adaptive metrics Ori-Adapt and FoR-Adapt combine both scenarios to examine the model's overall ability to exhibit contextually appropriate spatial reasoning based on object properties, providing a comprehensive assessment of spatial reasoning alignment with human cognition strategies.

## B  EVALUATION DETAILS

### B.1  HUMAN EVALUATION

We evaluate human performance on the SOFA benchmark with 6 PhD students from an NLP research laboratory. Humans achieve 100% and 92% accuracy on orientation and spatial relationship tasks respectively establishing performance upper bounds for model comparison. Prior works (Ziegler et al., 2012; Johannsen & de Ruiter, 2013; Dobnik et al., 2014) has shown experimental results that humans prefer OC frames in spatial communication. Humans adopt this strategy particularly for fronted objects with clear directional cues to resolve ambiguity. Our human evaluation confirms these findings empirically and validates our benchmark quality. The controlled nature of our synthetic scenes allows for more precise human judgments. The near-perfect human performance also validates our object selection, confirming that the chosen objects indeed possess clear frontedness.

### B.2  EVALUATION SETUP

We evaluate all VLMs using consistent inference settings with temperature set to 0.1 and max tokens to 512, with batch size of 1 for all models. Large models (70B+ parameters) require 4 NVIDIA A6000 GPUs (48GB VRAM) for inference, while smaller models ($\leq$13B parameters) run on a single A6000 GPU, and proprietary models (GPT-4o, Gemini-2.0-Flash) are accessed via APIs. Since all evaluated models are instruction-tuned and generally follow the specified answer format, we extract responses from the final answer portion using exact matching. The object categories used for matching are the four figure objects in each scene: vase, lamp, dumbbell, and table. For orientation tasks involving non-fronted objects, we accept various forms of negative responses including "No", "None", and similar variations indicating absence of orientation. Each model is evaluated once on the SOFA benchmark, with accuracy calculated based on whether predictions match the ground truth object categories.

## C  TRAINING DETAILS

### C.1  RELATIONAL ANNOTATION

Our approach aims to teach VLMs object-centric spatial reasoning grounded in object properties and inter-object relationships. With controlled synthetic scenes where objects serve as spatial landmarks aligned with coordinate axes, we construct relational annotations that teach models abstract spatial concepts. This design is inspired by human use of landmarks to encode surrounding environments in mental representations (Tversky, 2003), with intrinsically oriented objects providing directional cues for reference frames (Mou & McNamara, 2002; Li et al., 2011). For orientation understanding, we avoid directional terms that could be confused with camera perspectives. Instead, we describe orientations through inter-object relationships such as "the sofa faces the table" where the table serves as a landmark positioned along the sofa's intrinsic front. For spatial relationship annotation, we eliminate FoR-related ambiguity by systematically applying contextually appropriate reference frames based on object properties. We annotate spatial relations using OC frames for fronted objects and VC frames for non-fronted objects.

### C.2  OBJECT CATEGORY SELECTION

Our training dataset contains 47,232 images with 283,392 image-text pairs, providing approximately 6 queries per image (2 orientation queries and 4 spatial relationship queries). From the

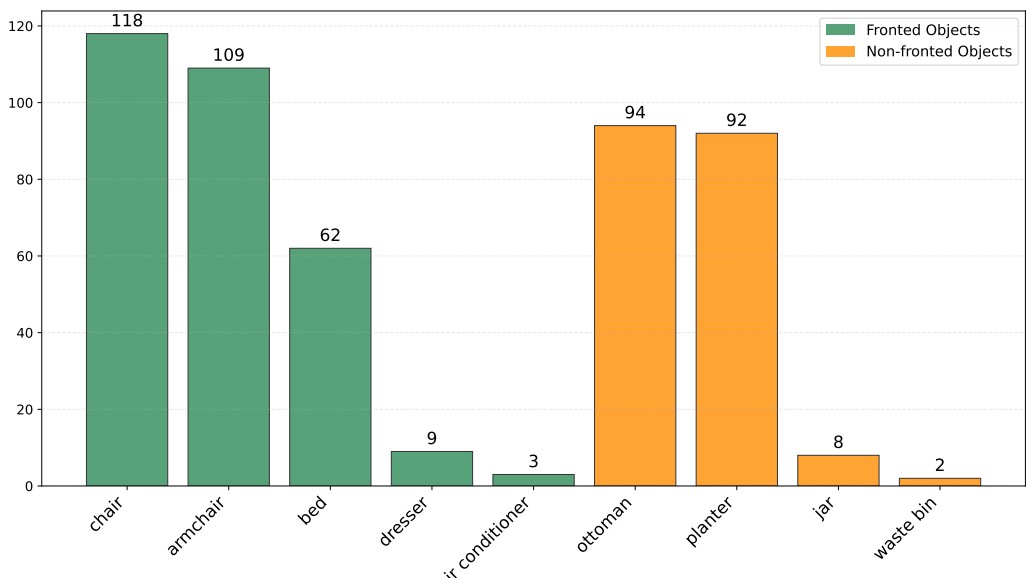

Figure 9: Distribution of object categories in the training dataset. The dataset contains 9 categories with 499 total instances, excluding any object meshes used in the SOFA benchmark. Green bars represent fronted objects, while orange bars represent non-fronted objects.

ABO dataset (Collins et al., 2022), we deliberately select ground objects based on the clarity of their frontedness to organize synthetic spatial scene. Our selection comprises 9 object categories with 499 total instances: 5 fronted categories and 4 non-fronted categories. Fronted objects (301 instances) possess inherent directional cues for spatial reasoning, while non-fronted objects (198 instances) lack such intrinsic directionality. To ensure fair evaluation, we exclude any object meshes that appear in our benchmark SOFA. The detailed distribution of object categories is shown in Figure 9.

## C.3 TRAINING CONFIGURATION

We fine-tune `Qwen2.5-VL-7B-Instruct`[2] and `llama3-llava-next-8b-hf`[3] on our object-centric spatial training dataset. We apply LoRA (Hu et al., 2022) to preserve existing spatial reasoning capabilities, which utilize camera viewpoints as primary origin for spatial descriptions. During the training, we unfreeze the vision encoder to capture fine-grained visual features for distinguishing between orientational states and recognizing object frontedness. We implement LoRA adaptation with rank=64, alpha=128, and dropout=0.05, applying it to the LLM component's attention and feed-forward layers (`q_proj`, `k_proj`, `v_proj`, `o_proj`, `gate_proj`, `up_proj`, `down_proj`). Training is conducted for 3 epochs with a learning rate of 1e-4, using a batch size of 16 per device across 4 NVIDIA A6000 GPUs (48GB VRAM) with gradient accumulation steps of 2, resulting in a global batch size of 128. We select Qwen2.5-VL-7B-Instruct as our primary model for comprehensive analysis, due to its balanced performance across both tasks. For ablation studies, we train individual task models using identical training configurations.

## D  REAL-WORLD SCENARIOS

### D.1  REAL-WORLD DIVERSITY

While our training dataset focused on abstract spatial concept learning using four landmarks (lamp, table, dumbbell, vase) as figure objects, the real-world evaluation encompasses a significantly broader range of object categories to assess generalization capabilities. The real-world dataset contains 280 queries across 54 distinct figure object categories, demonstrating the model's ability to handle previously unseen objects. These categories include both overlapping objects from training

---

[2] https://huggingface.co/Qwen/Qwen2.5-VL-7B-Instructt
[3] https://huggingface.co/llava-hf/llama3-llava-next-8b-hf

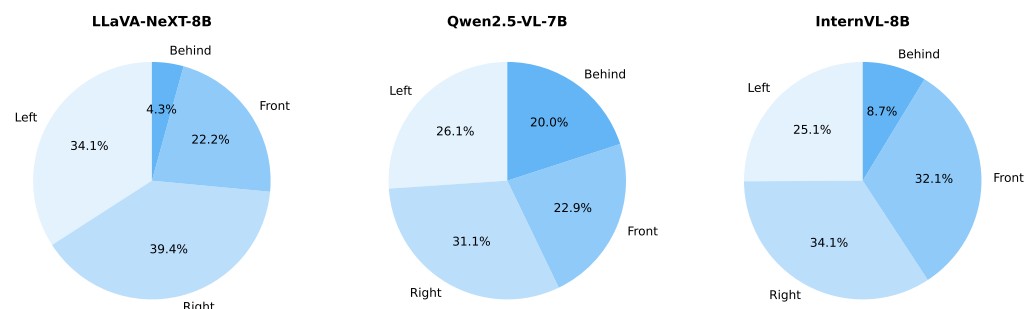

Figure 10: Model prediction distribution on real-world spatial relation queries. LLaVA-NeXT shows extreme lateral bias (left/right), Qwen demonstrates the most balanced predictions, while InternVL favors front and right directions but under-predicts behind relations.

and entirely novel categories such as television, christmas tree, laptop, door, piano, refrigerator, and various household items. Additionally, unlike the controlled synthetic scenes where ground objects are consistently center-positioned with canonical orientations, real-world images feature ground objects in arbitrary positions and rotations, requiring more robust spatial understanding. This diversity tests the model's capacity to apply abstract spatial concepts beyond the controlled synthetic environment to realistic indoor scenes with natural object arrangements and spatial distributions.

### D.2 PREDICTION PATTERNS

As shown in Figure 10, Examination of prediction distributions on real-world images reveals systematic biases across baseline models. LLaVA-NeXT shows extreme lateral bias with left/right predictions accounting for 73.5% of responses, while barely predicting "behind" relations (4.3%). This pattern reflects the label imbalances in LLaVA training datasets where left/right relationships dominate spatial relation distributions (Chen et al., 2025). Qwen demonstrates the most balanced distribution with more frequent "behind" predictions (20%). InternVL shows preference for "front" (32.1%) and "right" (34.1%) relations but still under-predicts "behind" (8.7%). These prediction patterns help explain the observed performance differences across models on different spatial relations.

## E LIMITATION AND FUTURE WORKS

**Spatial Modeling Constraints.** We limit our scope to discrete 90° orientation increments and basic directional terms (front, behind, left, right) for controlled evaluation, without addressing fine-grained orientations and richer spatial expressions such as distance-based relations (near, far). Additionally, we do not account for scale dependency–how object size differences or inter-object distances influence spatial reasoning–nor handle complex multi-object spatial hierarchies or compositional scene understanding. Future work should extend toward continuous orientation angles, comprehensive spatial vocabulary, and multi-scale spatial modeling to address these limitations.

**Physical Embodiment Applications.** A particularly promising direction lies in embodied AI applications where spatial understanding must inform physical actions. Integrating our object-centric spatial reasoning framework with vision-language-action systems could enable more effective robotic manipulation and navigation tasks. This would bridge the gap between visual-linguistic spatial understanding and real-world physical interaction, opening new possibilities for human-robot spatial communication and collaboration.

## F LLM USAGE

We used a large language model (LLM) as a writing assistant for polish purposes, including improving clarity of explanations, refining sentence structure, and organizing content presentation. All core ideas, experimental design, and results analysis were conducted by the authors.

