# OpenReview forum: "Aligning Vision-Language Models With Human Directional Reference"
_ICLR.cc/2026/Conference — ICLR 2026 Conference Withdrawn Submission_

### Official Review · Reviewer_gpD2 · 2025-10-24

**Soundness:** 3
**Presentation:** 3
**Contribution:** 1
**Rating:** 2
**Confidence:** 4

**Summary:**

Spatial expressions are ambiguous due to perspective taking. This paper mainly studies the problem of object-centered spatial reasoning that associate with objects that has an intrinsic front. This paper introduces a new synthetic benchmark to systematically evaluate spatial reasoning of vision-language models (VLMs) and experiments show that VLMs are not good at identify object orientations and use a view centric perspective. They also show that spatial reasoning from an object centric perspective achieves better alignment with human behavior.

**Strengths:**

1. Well-design figures show clearly of the spatial reasoning problem being studied. The paper writing is also easy to understand with good flows.
2. The ablation results show the importance of both orientation prediction and object centric spatial reasoning.

**Weaknesses:**

The contribution in this paper is very limited. The COMFORT paper referenced has studied the problem of perspective taking, object frontness, and showed that vision language models (VLMs) adopt a view-centric perspective and cannot flexibly use other perspectives like object-centric perspective (i.e., intrinsic FoR). Therefore the only contribution is showing that fine-tuning VLMs for object-centric spatial reasoning lead to better performance, but it's not very surprising for this finding.

**Questions:**

1. For human evaluation, participants are from 6 PhD students from an NLP research laboratory. Is this standard for participant recruiting? It lacks of diversity of the participants and PhD students in an NLP background maybe familiar with the spatial reasoning problems and will make the participants not representative for human performance.
2. It's better to bold or underscore the best model performance and second best performance in the tables to make the results easier to compare.

---

> ### Author Response · Authors · 2025-11-20
> **Response to Reviewer gpD2**
>
> ### Response to weakness:
>
> ### W1 . Clarifying the Novel Contributions Beyond Prior Work
>
> We appreciate the reviewer’s comment. Our focus is on **how humans resolve FoR ambiguity through frontedness-based adaptive selection** and whether VLMs can achieve this human-like mechanism. Our contributions are:
>
> 1. **Frontedness-driven adaptive selection**: We identify frontedness as the key mechanism—humans adopt object-centric frames for fronted objects and viewer-centric frames for non-fronted objects—and systematically evaluate whether VLMs understand this strategy.
> 2. **Diagnostic decomposition**: Our three-component breakdown (frontedness recognition → orientation understanding → intrinsic axial structure) systematically reveals **where and why** VLMs fail, going beyond observing that they "cannot flexibly use object-centric perspective."
> 3. **Generalization through abstract spatial learning**: Fine-tuning on synthetic scenes with abstract spatial concepts learned through inter-object relationships transfers successfully to real images, demonstrating genuine spatial understanding rather than scene memorization.
>
> Comparing to COMFORT, which focuses on identifying the selected FoR, our work reveals the mechanism that drives these choices and the specific component responsible for object-centric FoR failures.
>
> &nbsp;
> ### Response to questions:
>
> ### Q1. Human-Study Representativenes
>
> > For human evaluation, participants are from 6 PhD students from an NLP research laboratory. Is this standard for participant recruiting? It lacks of diversity of the participants and PhD students in an NLP background maybe familiar with the spatial reasoning problems and will make the participants not representative for human performance.
> >
>
> We appreciate this concern. Literature survey on spatial reference frame selection (e.g.; Burigo & Sacchi, 2013) conducted similar human evaluations with diverse age groups and larger sample sizes, showing consistent results with our findings. Our human evaluation serves to validate data quality and confirm that the frontedness-driven FoR selection mechanism aligns with established human behavior patterns.
>
> &nbsp;
> ### Q2. Table Clarity Suggestion
>
> > It's better to bold or underscore the best model performance and second best performance in the tables to make the results easier to compare.
> >
>
> Thank you for the suggestion. We will bold the best performance and underline the second-best in the revised version.

---

> > ### Comment · Reviewer_gpD2 · 2025-11-20
> >
> > Thanks for your clarifications. I acknowledged diagnostic decomposition and generalization through abstract spatial learning so I have raised my score, but the novelty part is still limited. For example, the frontness-based adaptive selection (i.e., object-centered intrinsic FoR) is explicitly studied in COMFORT:
> >
> > 1. "The relative FoR positions a viewer (egocentric or addressee) as the origin, focusing on the observer’s intrinsic perspective. Liu et al. (2010) have highlighted the ambiguities in situated communication among three variations of intrinsic and relative FoRs (Figure 2): the egocentric relative, the addressee-centered relative, and the object-centered intrinsic FoRs."
> >
> > 2. "When the relatum is fronted (e.g., Figure 1a), multiple FoRs can be explicitly adopted to interpret the scene. A COMFORT-CAR image, therefore, involves the egocentric perception of a referent, a fronted relatum, and an additional human addressee. One can interpret the spatial relations using either the Camera, Addressee, or Relatum (C/A/R) as the origin to resolve the reference frame ambiguity."
> >
> > 3. "We find that all models, including the strong ones like GPT-4o and InternLM-XComposer2, show close-to-chance performance (50% accuracy) when being prompted to use the intrinsic or addressee-centered relative FoRs."

---

> > > ### Author Response · Authors · 2025-11-24
> > > **Response to Reviewer Comments**
> > >
> > > Thank you for taking the time to engage further with our work and for your thoughtful feedback. We appreciate your recognition of the diagnostic decomposition and generalization through abstract spatial learning. We would like to highlight that the key difference:  COMFORT observes which FoR models select in ambiguous scenarios; our work investigates whether VLMs can learn the human strategy of adaptively selecting object-centric FoR based on frontedness and intrinsic axial structure to resolve such ambiguity.
> > >
> > > - While COMFORT examines which FoR a model selects in ambiguous scenes (Camera/Addressee/Relatum), **it does not test whether models can perceive frontedness itself,** and only reports model choices in left/right-facing fronted scenes rather than evaluating broader orientation understanding.
> > > - Our work identifies **why models fail**—they lack frontedness recognition and orientation understanding—and demonstrates that with targeted training on abstract spatial concepts, models can **acquire the human strategy** of adaptively selecting object-centric FoR based on frontedness to resolve ambiguity across different viewpoints in communication.
> > >
> > > We sincerely hope that this clarification helps convey the distinctions and contributions of our work.

---

> > > > ### Comment · Reviewer_gpD2 · 2025-11-26
> > > >
> > > > I acknowledged some novel settings but they are quite limited. Additionally, due to the reason that fine-tuning leading to better results in both synthetic tasks and real-world tasks is not surprising and has been studied in many papers like [1], I maintain my current rating.
> > > >
> > > > [1] Ray, Arijit, et al. "SAT: Dynamic Spatial Aptitude Training for Multimodal Language Models." arXiv preprint arXiv:2412.07755 (2024).

---

> > > > > ### Author Response · Authors · 2025-12-02
> > > > > **Response to Reviewer Comments**
> > > > >
> > > > > We thank the reviewer for the continued discussion. We would like to clarify a key distinction regarding our fine-tuning approach:
> > > > >
> > > > > - **Conceptual learning through abstract spatial structure**
> > > > >
> > > > > Our approach differs fundamentally from simulation-based methods. Spatial terms like "left/right/front/behind" are abstract relational concepts. **To learn these concepts, models must internalize abstract spatial structure—understanding spatial relationships through inter-object configurations rather than memorizing individual examples.**
> > > > >
> > > > > We use synthetic scenes to systematically present these abstract spatial relationships, allowing models to acquire generalizable spatial reasoning capability. **The key insight is that internalizing abstract spatial structure is necessary for transferable spatial reasoning.** Fine-tuned models demonstrate strong performance not only on synthetic tasks but also in real-world scenarios, showing effective transfer of abstract spatial concepts.

---

### Official Review · Reviewer_eTNt · 2025-10-31

**Soundness:** 3
**Presentation:** 3
**Contribution:** 2
**Rating:** 4
**Confidence:** 5

**Summary:**

The paper formalizes object-centered (OC) spatial reasoning through frontedness (intrinsic "front" of objects) and introduces SOFA, a synthetic 3D benchmark that tests (i) object orientation understanding and (ii) spatial relation reasoning with appropriate frame-of-reference (FoR) selection (OC for fronted objects, viewer-centered/VC for non-fronted).

Baseline VLMs show canonical/front-facing bias and VC bias. After task-specific fine-tuning (LoRA + unfrozen vision encoder) on synthetic relational annotations, two mid-size VLMs approach human-like FoR selection and improve orientation accuracy, with modest transfer to a small real-image slice. The work is conceptually clear and methodologically careful, but several wins appear to stem from in-domain synthetic training aligned tightly with the eval, so the leap beyond the trained domain is still tentative.

**Strengths:**

- The key insight is making it clear when to use object-centered vs viewer-centered reasoning. This makes the model's competence auditable, not just whether it gets "left/right" correct. That separation is the paper's real conceptual contribution. The paper quantifies two specific biases in modern VLMs: they assume objects are in canonical/front-facing poses and default to viewer-centric perspectives. This makes "VC bias" a measurable phenomenon rather than just anecdotal observation.
- Joint training is synergistic. Training on orientation alone helps with orientation but not object-centered relations. Training on OC relations alone helps with relations and somewhat helps orientation. Training on both together works best, suggesting these skills mutually reinforce each other. The approach uses LoRA on the language model while unfreezing the vision encoder, trained on synthetic data with combined losses for both orientation and OC reasoning. After fine-tuning, models show large improvements on SOFA benchmarks, getting much closer to human performance. The ablations confirm that you really need both types of training together.
- "Real" transferable. On a small MS-COCO subset focused on fronted objects, the fine-tuned models show some transfer to real images, though performance is still well below human level. This suggests the approach has promise beyond synthetic data, even if there's a long way to go.

**Weaknesses:**

- Data quality: Very obviously, the data suffer from uniform lighting, canonical placements, and 90° rotations do not capture continuous headings, occlusion, or clutter.
- The strongest gains reported indeed stem from fine-tuning on a synthetic dataset (SOFA-train) that is structurally identical to the evaluation setting. In other words: the training and test data share identical rendering pipelines, spatial layouts (4-way axial placements, 90° rotations), and categorical frontedness priors; although the authors exclude test meshes, the category distribution and geometric conventions are nearly identical; the fine-tuned model directly optimizes for the same question templates and answer formats used in evaluation. So **this is largely "train in-domain, test in-domain" fine-tuning success rather than emergent, generalized spatial understanding.**
- Prior work such as SpaRE (https://arxiv.org/abs/2504.20648; ACL 2025) has already shown that targeted synthetic spatial data can boost VLM spatial reasoning, so using more synthetic supervision in a matched domain is not, by itself, novel. Without demonstrating cross-domain generalization (to real images, continuous viewpoints, or unseen object categories), the work risks being seen as an exercise in careful dataset engineering rather than a meaningful advance in spatial cognition or model alignment.

**Questions:**

### Q1: Are results robust to non-axis-aligned figure placements and continuous headings (not just 0, 90, 180, 270)?

**Action:** Add a continuous-angle split (e.g., every 15°), random radial offsets, and non-planar placements; track Ori/FoR metrics vs angle.

### Q2: Are models simply overfitting the prompt schema or a constrained label set?

**Action:** Swap templates (no "reasoning-then-answer"), change answer format (free-text vs object name), and test robustness to synonyms/extra distractors; include a "no-template" ablation.

### Q3: Does improvement hinge on category overlap (fronted vs non-fronted priors)?

**Action:** Create category-disjoint and appearance-disjoint splits (including novel fronted categories) and report Ori/FoR deltas; increase the real-image slice beyond 280 and include non-fronted real objects.

---

> ### Author Response · Authors · 2025-11-20
> **Response to Reviewer eTNt**
>
> ### Response to questions:
> ### Q1. Robustness Beyond Axis-Aligned Scenes
>
> > Are results robust to non-axis-aligned figure placements and continuous headings (not just 0, 90, 180, 270)?
> >
>
> > **Action:** Add a continuous-angle split (e.g., every 15°), random radial offsets, and non-planar placements; track Ori/FoR metrics vs angle
> >
>
> Thank you for the insightful suggestion. We prepared two evaluation scenarios with continuous angles and varied placements. We use 30° increments to avoid ambiguous directions (e.g., 45° creates ambiguity between front and left):
>
> 1. **Ground rotation (±30°)**: 5-object arrangements (1 ground, 4 figures) where ground objects are rotated in 30° increments. Questions follow the format “Related to {ground}, which object is {directional phrase} it?”
> 2. **Figure placement (±30°)**: 2-object arrangements (1 ground, 1 figure) where figure objects are placed at ±30° offsets from axis-aligned positions. Since only one figure is present, questions follow the format "Where is {figure} relative to {ground}?" with directional terms (front/back/left/right) as ground truth.
>
> |  | **Fronted** |  | **Nonfronted** |  |
> | --- | --- | --- | --- | --- |
> | **Model** | **Ori-F** | **FoR-OC** | **Ori-NF** | **FoR-VC** |
> | Qwen2.5-VL-7B | 0.197 | 0.053 | 0.604 | 0.776 |
> | Ours | 0.839 | 0.727 | 0.989 | 0.904 |
> | Ground rotation (±30°) | 0.617 | 0.627 | - | - |
> | Figure placement (±30°) | 0.736 | 0.575 | 0.802 | 0.792 |
>
> Our model maintains strong performance across continuous angles and diverse spatial placements, demonstrating robustness beyond axis-aligned configurations. We acknowledge that our synthetic evaluation focuses on planar configurations to isolate intrinsic axial structure understanding. However, our real image evaluation includes naturally occurring non-planar arrangements, showing that the learned spatial reasoning transfers to more complex 3D scenarios.
>
> &nbsp;
> ### Q2. Prompt/Label Overfitting
>
> > Are models simply overfitting the prompt schema or a constrained label set?
> >
>
> > **Action:** Swap templates (no "reasoning-then-answer"), change answer format (free-text vs object name), and test robustness to synonyms/extra distractors; include a "no-template" ablation.
> >
>
> We appreciate the reviewer for this concern. To verify that models learn genuine spatial understanding rather than overfitting prompt schemas, we tested two variations:
>
> 1. **w/o Template**: Removed reasoning-then-answer structure and simplified question format
> 2. **Direction term answers**: Changed answer format to directional terms for both orientation ("Which direction is {ground} facing?") and spatial relations ("Where is {figure} relative to {ground}?")
>
> |  | **Fronted** |  | **Nonfronted** |  |
> | --- | --- | --- | --- | --- |
> | **Model** | **Ori-F** | **FoR-OC** | **Ori-NF** | **FoR-VC** |
> | Qwen2.5-VL-7B | 0.197 | 0.053 | 0.604 | 0.776 |
> | Ours | 0.839 | 0.727 | 0.989 | 0.904 |
> | w/o Template | 0.734 | 0.675 | 0.844 | 0.94 |
> | Direction term | 0.806 | 0.714 | 0.989 | 0.927 |
>
> Performance remains strong across format variations, demonstrating robustness to prompt schemas and label formats. The model maintains spatial reasoning capabilities rather than memorizing specific templates.
>
> &nbsp;
> ### Q3. Category-Overlap/Scene-Memorization
>
> > Does improvement hinge on category overlap (fronted vs non-fronted priors)?
> >
>
> > **Action:** Create category-disjoint and appearance-disjoint splits (including novel fronted categories) and report Ori/FoR deltas; increase the real-image slice beyond 280 and include non-fronted real objects.
> >
>
>
> We thank the reviewer for this important concern. To verify that improvements are not driven by category overlap, we created category-disjoint and appearance-disjoint splits and expanded our real-image evaluation from 280 to 560 instances with 60 unique ground objects and 102 unique figure objects. This real-world evaluation includes diverse spatial placements, varied lighting conditions, and both fronted and non-fronted objects in naturalistic arrangements.
>
> **Synthetic split-based evaluation**:
>
> |  | **Ori-F** | **FoR-OC** | **Ori-NF** | **FoR-VC** |
> | --- | --- | --- | --- | --- |
> | Category-disjoint | 0.771 | 0.667 | 0.989 | 0.875 |
> | Appearnce-disjoint | 0.906 | 0.786 | 0.989 | 0.932 |
>
> **Real-image evaluation:**
>
> | **Model** | **FoR-VC** | **FoR-OC** | **FoR-Adapt** |
> | --- | --- | --- | --- |
> | LLaVA-NeXT-8B | 0.521 | 0.164 | 0.343 |
> | InternVL3-8B | 0.689 | 0.225 | 0.457 |
> | Qwen2.5-VL-7B | 0.482 | 0.156 | 0.319 |
> | Ours | 0.657 | 0.464 | 0.561 |
>
> Results demonstrate that our model generalizes beyond category priors and controlled synthetic settings, achieving strong performance on novel categories and maintaining superior adaptive FoR selection across real-world complexity.

---

> > ### Comment · Reviewer_eTNt · 2025-11-21
> >
> > Thank you for the careful rebuttal!
> >
> > I think my concerns about the Questions part are clear. However, the authors did not get me back on the Weaknesses I raised. Specifically, my concern is with Weakness 2, that this work can be interpreted as a "train in-domain, test in-domain" fine-tuning success. If authors can provide me a satisfying response on this point, I would consider raising my ratings accordingly.

---

> > > ### Author Response · Authors · 2025-11-24
> > > **Clarifications Regarding Weaknesses**
> > >
> > > Thank you for your thoughtful review and constructive feedback. We greatly appreciate your careful evaluation and the opportunity to address your concerns. We apologize for not fully addressing the weaknesses in our previous response; we had initially understood that some of these points were addressed in the Questions section. We now provide a focused clarification to ensure that your concerns are thoroughly addressed.
> > >
> > > &nbsp;
> > > ### Reponse to Weakness:
> > >
> > > ### W1. Limited scene variation and synthetic simplicity
> > >
> > > > Data quality: Very obviously, the data suffer from uniform lighting, canonical placements, and 90° rotations do not capture continuous headings, occlusion, or clutter.
> > > >
> > >
> > > We understand the concern regarding SOFA's synthetic scenes with canonical placements and discrete rotations. While real-world scenarios exhibit continuous and non-axis-aligned poses, **we first needed a systematic evaluation framework to diagnose whether models rely on camera coordinates (viewer-centric) or object intrinsic axes (object-centric)**. Aligning object axes with camera axes is essential—continuous angles would introduce ambiguity in attributing spatial descriptions to a specific FoR. These controlled conditions isolate the cognitive mechanisms—frontedness recognition and adaptive FoR selection—without confounds from lighting, clutter, or pose ambiguity. To address limitations of controlled settings, we also evaluated on real images with natural clutter, lighting variation, and continuous poses.
> > >
> > > &nbsp;
> > > ### W2. In-domain train/test and concerns about generalization
> > >
> > > > The strongest gains reported indeed stem from fine-tuning on a synthetic dataset (SOFA-train) that is structurally identical to the evaluation setting. In other words: the training and test data share identical rendering pipelines, spatial layouts, and categorical frontedness priors; although the authors exclude test meshes, the category distribution and geometric conventions are nearly identical; the fine-tuned model directly optimizes for the same question templates and answer formats used in evaluation. So **this is largely "train in-domain, test in-domain" fine-tuning success rather than emergent, generalized spatial understanding.**
> > > >
> > >
> > > Thank you for this important point. To address generalization, we tested on synthetic benchmarks with unseen object categories and on real images with diverse categories and question formats—verifying spatial reasoning beyond specific training conditions. As you suggested, we conducted additional evaluations to assess robustness:
> > >
> > > - **Continuous-angle evaluation (Q1)**: Tested with continuous ground rotations and non-canonical placements in two-object configurations, demonstrating robustness beyond the training configuration
> > > - **Template variations (Q2)**: Ensured spatial reasoning was not tied to specific prompt formats
> > > - **Expanded real-image evaluation**: Leveraging the VSR[1] benchmark, we increased evaluation from 280 to 560 image-test pairs (60 unique ground objects, 102 figure objects) to further assess FoR-VC and FoR-Adapt
> > >
> > > These evaluations demonstrate that the model acquired **generalizable spatial reasoning mechanisms** rather than memorizing synthetic training patterns. Notably, robust performance on real images confirms learned spatial reasoning transfers beyond controlled synthetic settings. We will include the extended results in the revised manuscript with representative real-image examples demonstrating generalization beyond synthetic training conditions.
> > >
> > > &nbsp;
> > > ### W3. The limitations of synthetic supervision
> > >
> > > > Prior work such as SpaRE [2] has already shown that targeted synthetic spatial data can boost VLM spatial reasoning, so using more synthetic supervision in a matched domain is not, by itself, novel. Without demonstrating cross-domain generalization, the work risks being seen as an exercise in careful dataset engineering rather than a meaningful advance in spatial cognition or model alignment.
> > > >
> > >
> > > We appreciate this point. Prior work such as SpaRE [2] leveraged real images with synthetic spatial annotations. In contrast, our approach uses **controlled synthetic scenes** to teach **abstract spatial concepts**, specifically the **intrinsic axial structure** of objects, enabling frontedness recognition and adaptive FoR selection.
> > >
> > > To verify generalization beyond synthetic conditions, we evaluated the model on **real images** with:
> > >
> > > - diverse object categories
> > > - arbitrary placements and rotations
> > >
> > > Although trained on only **4 figure object categories** in synthetic scenes, the model successfully generalizes to **102 unique figure objects** in arbitrary real-world configurations, demonstrating that it learned the **underlying spatial reasoning mechanisms** rather than memorizing object- or configuration-specific patterns.
> > >
> > > ---
> > >
> > > [1] Visual spatial reasoning, Liu et al., TACL, 2023
> > >
> > > [2] SpaRE: Enhancing Spatial Reasoning in Vision‑Language Models with Synthetic Data, M. Ogezi and F. Shi, ACL, 2025.

---

> > > > ### Comment · Reviewer_eTNt · 2025-11-26
> > > >
> > > > Thanks for the response. My concern is that your rebuttal effectively follows the pattern:
> > > > - you are challenged on in-domain training/testing, ...
> > > > - you add some additional evaluations, and then you conclude that these “demonstrate generalizable spatial reasoning.”
> > > >
> > > > However,
> > > > - you do not clearly specify how these new evaluations are distributionally different from the training setup, ...
> > > > - you do not compare against strong controls (e.g., matched synthetic fine-tuning without FoR structure or with shuffled labels), ...
> > > > - you implicitly assume that robustness under modest perturbations implies a genuinely generalizable mechanism.
> > > >
> > > > This is exactly the point under dispute, so the rebuttal is logically circular: it asserts the existence of a generalizable mechanism rather than providing evidence that rules out in-domain pattern exploitation.

---

> > > > > ### Author Response · Authors · 2025-12-02
> > > > > **Response to Reviewer Comments**
> > > > >
> > > > > We thank the reviewer for the feedback. To clarify potential concerns about in-domain pattern exploitation, we designed four robustness evaluations introducing clear train–test distribution shifts. The table below summarizes model performance under each setting.
> > > > >
> > > > > | Evaluation Setting | **Fronted** |  | **Non-fronted** |  | **Adaptive** |  |
> > > > > | --- | --- | --- | --- | --- | --- | --- |
> > > > > |  | Ori-F | FoR-OC | Ori-NF | FoR-VC | Ori-Adapt | FoR-Adapt |
> > > > > | **Standard setting** | 0.839 | 0.727 | 0.989 | 0.904 | 0.869 | 0.771 |
> > > > > | **Geometric Shift** | 0.617 | 0.627 | – | – | – | – |
> > > > > | **Layout Shift**  | 0.736 | 0.575 | 0.802 | 0.792 | 0.749 | 0.629 |
> > > > > | **Linguistic Shift**  | 0.734 | 0.675 | 0.844 | 0.940 | 0.756 | 0.742 |
> > > > > | **Label-Space Shift**  | 0.806 | 0.714 | 0.989 | 0.927 | 0.843 | 0.767 |
> > > > > 1. **Geometric Shift — Continuous Ground Rotation (±30°)**
> > > > >
> > > > >     In the **Geometric Shift**, ground objects are rotated in continuous 30° increments rather than being aligned to camera axes (0°, 90°, 180°, 270°) as in training. This introduces intrinsic–camera misalignments not present during training. Despite this novel geometric configuration, the model maintains strong object-centric reasoning (Ori-F: 0.617, FoR-OC: 0.627), indicating it infers intrinsic object front rather than relying on axis-aligned memorization.
> > > > >
> > > > > 2. **Layout Shift — Off-Axis Figure Placement (±30°)**
> > > > >
> > > > >     For the **Layout Shift**, evaluation scenes contain only a single figure positioned at ±30° offsets from canonical axial slots, whereas training uses four-object scenes with figures in discrete axial placements. In addition, evaluation requires directional-term answers (“front”, “back”, “left”, “right”) instead of object names. The model performs robustly under these conditions (Fronted: Ori-F 0.736, FoR-OC 0.575; Non-fronted: Ori-NF 0.802, FoR-VC 0.792), demonstrating that it does not rely on scene memorization or training label distributions.
> > > > >
> > > > > 3. **Linguistic Shift — Template Removal**
> > > > >
> > > > >     The **Linguistic Shift** removes the reasoning-then-answer template used in training, presenting question-only prompts. Performance remains strong (Fronted: Ori-F 0.734, FoR-OC 0.675; Non-fronted: Ori-NF 0.844, FoR-VC 0.940), showing that the model’s spatial reasoning is independent of template-specific cues.
> > > > >
> > > > > 4. **Label-Space Shift — Directional-Term Output**
> > > > >
> > > > >     Finally, the **Label-Space Shift** replaces the object-name outputs used during training with directional-term outputs. The model maintains high performance (Fronted: Ori-F 0.806, FoR-OC 0.714; Non-fronted: Ori-NF 0.989, FoR-VC 0.927), indicating it generalizes beyond memorized label mappings and produces semantically meaningful directional judgments.
> > > > >
> > > > >
> > > > > We evaluated our model under settings that differ from the training setup, including continuous rotations, off-axis figure placements, template-free prompts, and directional-term outputs. Despite these distributional shifts, the model maintains strong orientation and FoR performance. These results indicate that it has learned generalizable spatial reasoning beyond memorizing in-domain patterns. We believe this directly addresses the reviewer’s concern regarding overfitting to the training setup, and we will incorporate these analyses into the revised manuscript.

---

### Official Review · Reviewer_sJqf · 2025-11-01

**Soundness:** 3
**Presentation:** 2
**Contribution:** 2
**Rating:** 4
**Confidence:** 4

**Summary:**

This paper studies how VLMs handle ambiguous spatial language, arguing that models should often use an object-centeric frame of reference (FoR) anchored to an object’s intrinsic frontedness rather than defaulting to the viewer perspective. The authors introduce a synthetic 3D benchmark, SOFA, to test two skills: (1) recognizing an object’s facing direction (orientation) when frontedness exists, and (2) choosing the correct frame (object-centric or viewer-centric) when answering spatial-relation questions. They show current VLMs frequently misidentify orientations and overwhelmingly default to the camera’s view; with targeted fine-tuning using SOFA-style annotations, models align much better with human behavior on both tasks.

**Strengths:**

* The key ideas are grounded in a rich literature in psychology and linguistic that study how linguistic descriptions affect human spatial cognition.

* The writing is clear and coherent, addressing critical cognitive bottleneck for VLMs that should be solved to achieve human-level spatial understanding.

* Provides a systematic analysis based on the new SOFA benchmark, showing that recent VLMs still exhibit viewpoint-related biases in spatial reasoning.

* Beyond proposing a benchmark, the paper demonstrates that fine-tuning VLMs (LLaVA-NeXT and Qwen2.5-VL) on SOFT leads to notable improvements in spatial reasoning.

**Weaknesses:**

Although the paper introduces a new benchmark SOFA, the technical novelty of both this benchmark and the takeaway from the failure analysis are questionable, in comparison with multiple previous studies. I outline the main concerns below.

* **Missing related work**:

  * **RoboSpatial [1]** makes a similar claim that VLMs struggle to reason in "object-centric" FoR, and present a benchmark to assess and fine-tune VLMs. This work should be cited and a further discussion seems necessary to clarify the novelty of SOFA.

  * **SITE [2]** also includes a "spatial orientation" subset that evaluates how VLMs handle object-centric FoR. Moreover, this benchmark consists of real images rather than synthetic ones. **OmniSpatial [3]** and **ViewSpatial-Bench [4]** also include perspective-taking tasks. Could the author elaborate on the advantage of synthetic setups in relation to such real benchmarks?

* The novelty relative to COMFORT [5] is unclear. Please clarify how this work differs from or extends COMFORT's claims and analysis on the FoR understanding of VLMs.

* Moreover, the statement that **"SOFA reveals limitations in their understanding of orientation and a tendency to adopt the viewer-centric perspective"** may not be a new finding, since the benchmarks listed above (and COMFORT) have extenstively discussed this limitation of VLMs.

* The claim that **"limited consideration of FoR undermines the consistency and reliability of such evaluations"** (line 133) is confusing. For instance, Yin et al. [6] include perspective-taking tasks, so it's unclear which aspect of FoR these previous work are lacking. Could the authors please elaborate more on the core missing aspect of these previous benchmarks?

* **"These approaches remain misaligned with human strategies for resolving FoR-related ambiguity, which rely on an object’s intrinsic orientation rather than inconsistent perspectives."** (line 140-141). This comment is unclear. What does it mean to rely on "inconsistent perspectives"? Is the intention to emphasize that humans can integrate information across multiple perspectives?

* Does the training dataset consist of similar synthetic scenes as in Figure 7? A concerning aspect on the training/evaluation on SOFA is that the overall scene layout would mostly overlap among training and evaluation data. In this case, is there any chance that VLMs memorize the scene patterns? Would there be a more constrained way to show that fine-tuned models are indeed "aligned" to human FoR, rather than memorizing scene structures?

---

[1] RoboSpatial: Teaching Spatial Understanding to 2D and 3D Vision-Language Models for Robotics, Song et al., CVPR 2025

[2] SITE: towards Spatial Intelligence Thorough Evaluation, Wang et al., ICCV 2025

[3] OmniSpatial: Towards Comprehensive Spatial Reasoning Benchmark for Vision Language Models, Jia et al., 2025

[4] ViewSpatial-Bench: Evaluating Multi-perspective Spatial Localization in Vision-Language Models, Li et al., 2025

[5] Do Vision-Language Models Represent Space and How? Evaluating Spatial Frame of Reference Under Ambiguities, Zhang et al., ICLR 2025

[6] Spatial Mental Modeling from Limited Views, Yin et al., 2025

**Questions:**

* Is SOFA a fully synthetic dataset? Is there a potential to extend the data synthesis pipeline to generate more realistic scenes? The current state of SOFA seems quite limited with a small number of categories and fixed scene configurations (e.g., layout, lighting, texture).

* In Section 4.3, the data preparation process for real-world scenarios seems quite hand-crafted and small in size. Have the authors considered showing generalization using existing real-image benchmarks (e.g., RoboSpatial [1], SITE [2]) instead?

**Details Of Ethics Concerns:**

No concerns.

---

> ### Author Response · Authors · 2025-11-20
> **Response to Reviewer sJqf (1/3)**
>
> ### Response to weaknesses:
>
> We sincerely thank the reviewer for the detailed comments. We understand the concern regarding the novelty relative to prior spatial reasoning benchmarks. To clarify, **SOFA’s central contribution is unified around one core idea**:
>
> SOFA is the **first benchmark** to operationalize frontedness as the mechanism humans use to adaptively select a Frame of Reference (FoR), and to decompose this capability into the three prerequisite components required for human-like FoR resolution.
>
> This single mechanism—**frontedness-driven FoR selection**—and its **three-stage cognitive decomposition** (frontedness recognition → orientation understanding → intrinsic axial structure) form the conceptual novelty of SOFA. Below, we summarize which components each benchmark evaluates:
>
> | Benchmark | Frontedness Recognition | Orientation Understanding | Intrinsic Axial Structure | Adaptive FoR Selection |
> | --- | --- | --- | --- | --- |
> | RoboSpatial [1] | x | x | o | x |
> | SITE [2] | x | o | x | x |
> | COMFORT [3] | x | x | x | o |
> | SOFA (ours) | o | o | o | o (based on frontedness) |
>
> Existing benchmarks evaluate certain aspects of spatial reasoning, but **none examine whether a model can decide when to adopt an object-centric FoR**, because *frontedness* is not parameterized. This is the key novel dimension SOFA introduces.
>
> &nbsp;
> ### W1. Missing related work
>
> > RoboSpatial [1] makes a similar claim that VLMs struggle to reason in "object-centric" FoR, and present a benchmark to assess and fine-tune VLMs. This work should be cited and a further discussion seems necessary to clarify the novelty of SOFA.
> >
>
> > SITE [2] also includes a "spatial orientation" subset that evaluates how VLMs handle object-centric FoR. Moreover, this benchmark consists of real images rather than synthetic ones. OmniSpatial [3] and ViewSpatial-Bench [4] also include perspective-taking tasks. Could the author elaborate on the advantage of synthetic setups in relation to such real benchmarks?
> >
>
> We thank the reviewer for pointing out these relevant works. We will include these citations and agree that a comparison is necessary for clarifying SOFA’s novelty.
>
> - Response to **RoboSpatial [1]**: RoboSpatial assesses object-centric FoR but does not examine whether models can **decide when to use it**, because frontedness is not parameterized. SOFA’s novelty is to explicitly measure whether VLMs learn the **frontedness-based conditional switch**.
> - Response to **SITE [2], OmniSpatial [4], ViewSpatial-Bench [5]**:  While real-image benchmarks provide ecological validity, they cannot systematically **manipulate the variables** required to test frontedness-driven FoR selection. Thus, SOFA uses synthetic setups only for isolating the cognitive primitives underlying FoR selection, and complements this with **real-image generalization tests** to ensure no scene memorization.
>
> &nbsp;
> ### W2. Novelty relative to COMFORT
>
> > The novelty relative to COMFORT [5] is unclear. Please clarify how this work differs from or extends COMFORT's claims and analysis on the FoR understanding of VLMs.
> >
>
>  We thank the reviewer for highlighting COMFORT [3] and fully agree that it is an important benchmark for analyzing FoR choices in complex scenarios. COMFORT evaluates which FoR a model selects when multiple possible origins are present. **SOFA further extends this direction by operationalizing frontedness as the trigger for FoR selection and by decomposing FoR understanding into its underlying cognitive components.** This allows us to examine why models fail to adopt object-centric FoRs—specifically, which prerequisite component breaks—rather than only observing which FoR they select.

---

> ### Author Response · Authors · 2025-11-20
> **Response to Reviewer sJqf (2/3)**
>
> ### W3. The concern about novelty of finding
>
> > Moreover, the statement that "SOFA reveals limitations in their understanding of orientation and a tendency to adopt the viewer-centric perspective" may not be a new finding, since the benchmarks listed above (and COMFORT) have extenstively discussed this limitation of VLMs.
> >
>
>  We appreciate the reviewer’s observation. Indeed, prior work has noted that VLMs default to viewer-centric perspectives. SOFA adds novelty by identifying why:
>
> 1. **Frontedness recognition failure**: VLMs cannot determine frontedness, the prerequisite for knowing when to adopt object-centric frames
> 2. **Canonical orientation bias**: Even for fronted objects, VLMs fail to understand orientation due to reliance on canonical viewpoints
> 3. **Intrinsic axial structure limitation**: Our ablation shows that even when orientation is understood, VLMs cannot grasp intrinsic axes required for object-centric reasoning
>
> This **causal decomposition** is new and allows us to diagnose the source of VLMs’ viewer-centric bias, not merely observe it. Compared to prior works [3], which evaluates which FoR a model chooses in multi-origin settings, **SOFA operationalizes *frontedness* as the mechanism that triggers FoR selection and decomposes this process into its cognitive prerequisites (frontedness → orientation → intrinsic axes)**. This enables causal analysis of why VLMs fail to adopt object-centric FoRs.
>
> &nbsp;
> ### W4. Limited FoR consideration
>
> > The claim that "limited consideration of FoR undermines the consistency and reliability of such evaluations" (line 133) is confusing. For instance, Yin et al. [6] include perspective-taking tasks, so it's unclear which aspect of FoR these previous work are lacking. Could the authors please elaborate more on the core missing aspect of these previous benchmarks?
> >
>
> We thank the reviewer for seeking clarification. Prior benchmarks including MINDCUBE [6] evaluate spatial reasoning across allocentric and egocentric viewpoints. However, without explicitly specifying the frame of reference, spatial annotations remain ambiguous—the same spatial relationship (e.g., "A is left of B") can be correct in viewer-centric frames but incorrect in object-centric frames, or vice versa. This ambiguity undermines reliable evaluation. Our statement intended to emphasize that **clear spatial expressions require explicit frame of reference specification.**
>
> &nbsp;
> ### W5. Inconsistent perspective
>
> > "These approaches remain misaligned with human strategies for resolving FoR-related ambiguity, which rely on an object’s intrinsic orientation rather than inconsistent perspectives." (line 140-141). This comment is unclear. What does it mean to rely on "inconsistent perspectives"? Is the intention to emphasize that humans can integrate information across multiple perspectives?
> >
>
> We apologize for the confusion. **By "inconsistent perspectives," we mean that viewer-centric frames produce different spatial descriptions for the same object arrangement depending on observer (camera) position** (See Fig. 1 for an illustration of how viewer-centric descriptions vary across perspectives while object-centric descriptions remain stable). While integrating information across multiple viewpoints is valuable, humans resolve this inconsistency by adopting object-centric frames for fronted objects, providing viewpoint-invariant descriptions. SOFA evaluates whether VLMs understand this frontedness-driven frame selection mechanism.
>
> &nbsp;
> ### W6. Scene memorization concern
>
> > Does the training dataset consist of similar synthetic scenes as in Figure 7? A concerning aspect on the training/evaluation on SOFA is that the overall scene layout would mostly overlap among training and evaluation data. In this case, is there any chance that VLMs memorize the scene patterns? Would there be a more constrained way to show that fine-tuned models are indeed "aligned" to human FoR, rather than memorizing scene structures?
> >
>
> We thank the reviewer for raising this concern. To ensure models learn generalizable spatial reasoning rather than memorizing scene patterns, we address this through real image evaluation. **Unlike the training dataset with fixed arrangements and 4 figure objects**, t**he real image evaluation uses (1) arbitrary spatial arrangements and (2) diverse objects**, showing that fine-tuned models perform well across all spatial relations.

---

> > ### Author Response · Authors · 2025-11-20
> > **Response to Reviewer sJqf (3/3)**
> >
> > ### Response to questions:
> >
> > ### Q1. Synthetic dataset with limited variation
> >
> > > Is SOFA a fully synthetic dataset? Is there a potential to extend the data synthesis pipeline to generate more realistic scenes? The current state of SOFA seems quite limited with a small number of categories and fixed scene configurations (e.g., layout, lighting, texture).
> > >
> >
> > Yes, SOFA is fully synthetic, and the pipeline can be extended to generate more categories, layouts, textures, and lighting conditions. The synthetic scenes allow controlled evaluation of FoR-specific reasoning. Generalization is confirmed on real images with **arbitrary layouts** and **diverse object categories**, showing that models apply spatial concepts beyond the synthetic setup. This demonstrates robustness to varied and realistic scenarios.
> >
> > &nbsp;
> > ### Q2. Small real-world set and choice of existing benchmarks
> >
> > > In Section 4.3, the data preparation process for real-world scenarios seems quite hand-crafted and small in size. Have the authors considered showing generalization using existing real-image benchmarks (e.g., RoboSpatial [1], SITE [2]) instead?
> > >
> >
> > We thank the reviewer for this suggestion. To demonstrate generalization and verify viewer-centric reasoning, we extended our real-world evaluation by including VSR[7], a subset of SITE [2]. We filtered VSR to include only the four prototypical directional terms (front/back/left/right) and images with identifiable frontedness, resulting in 560 evaluation instances.
> >
> > | **Model** | **FoR-VC** | **FoR-OC** | **FoR-Adapt** |
> > | --- | --- | --- | --- |
> > | **LLaVA-NeXT-8B** | 0.521 | 0.164 | 0.343 |
> > | **InternVL3-8B** | 0.689 | 0.225 | 0.457 |
> > | **Qwen2.5-VL-7B** | 0.482 | 0.156 | 0.319 |
> > | **Ours** | 0.657 | 0.464 | 0.561 |
> >
> > The results show that our model performs well on both viewer-centric and object-centric reasoning, adaptively selecting appropriate frames across real-world scenarios.
> >
> > ---
> >
> > [1] RoboSpatial: Teaching Spatial Understanding to 2D and 3D Vision-Language Models for Robotics, Song et al., CVPR 2025
> >
> > [2] SITE: towards Spatial Intelligence Thorough Evaluation, Wang et al., ICCV 2025
> >
> > [3] Do Vision-Language Models Represent Space and How? Evaluating Spatial Frame of Reference Under Ambiguities, Zhang et al., ICLR 2025
> >
> > [4] OmniSpatial: Towards Comprehensive Spatial Reasoning Benchmark for Vision Language Models, Jia et al., 2025
> >
> > [5] ViewSpatial-Bench: Evaluating Multi-perspective Spatial Localization in Vision-Language Models, Li et al., 2025
> >
> > [6] Spatial Mental Modeling from Limited Views, Yin et al., 2025
> >
> > [7] Visual spatial reasoning, Liu et al., TACL, 2023

---

> > > ### Author Response · Authors · 2025-11-26
> > > **Gentle Reminder**
> > >
> > > **Dear Reviewer sJqf,**
> > >
> > > Thank you for your time and effort in reviewing our paper. We sincerely appreciate your valuable feedback and have carefully considered all the points you raised.
> > >
> > > During this discussion period, we wanted to reach out to see if our responses have sufficiently addressed your concerns. We are happy to discuss any aspects of the paper that require further clarification or additional explanation—please let us know if there are any points you would like us to elaborate on further.
> > >
> > > Once again, we greatly appreciate your thoughtful review and insights.

---

> > > > ### Comment · Reviewer_sJqf · 2025-11-27
> > > >
> > > > I thank the authors for their detailed response to each point. However, there are still several issues that remain unclear to me.
> > > >
> > > > **Regarding W2 and W3**
> > > >
> > > > I still find it difficult to agree that these points offer significant technical novelty relative to COMFORT:
> > > >
> > > > * “operationalizing frontedness as the trigger for FoR selection”
> > > >
> > > > * “decomposing FoR understanding into its underlying cognitive components”
> > > >
> > > > First, identifying **”frontedness”** as a key component of FoR selection was already discussed in COMFORT and in prior work such as ViewSpatial-Bench and RoboSpatial. Although those works primarily use the term **”orientation”**, it appears to refer to the same (or a closely related) concept that the authors highlight as novel. That is, it is unclear to me why focusing on “frontedness” should be considered a new contribution in comparison to the previously discussed notion of “object orientation.”
> > > >
> > > > Second, I’m unsure whether the proposed causal decomposition of FoR selection for humans can or should be directly applied to assess the inference process of VLMs. The fact that human cognitive processing may follow a sequence “A→B→C” doesn’t imply that VLMs must follow the same sequence. Moreover, this alone does not seem sufficient to conclude that a particular step in that sequence is the root cause of failure in VLMs.
> > > >
> > > > Overall, I believe there are still unresolved ambiguities regarding the novelty of both the experimental approach and the findings. Therefore, I maintain my original score.

---

> > > > > ### Author Response · Authors · 2025-12-02
> > > > > **Response to Reviewer Comments**
> > > > >
> > > > > Thank you very much for your thoughtful feedback. We would like to clarify two key points regarding novelty and the role of frontedness in FoR understanding.
> > > > >
> > > > > 1. **Frontedness is not equivalent to orientation.**
> > > > >
> > > > >     Orientation is only meaningful for objects that have a determinate front. Existing benchmarks evaluate orientation but do not assess whether models can detect when frontedness is absent. This is a critical gap: **without recognizing frontedness, a model cannot meaningfully determine orientation** and therefore cannot know when to adopt an object-centric frame.
> > > > >
> > > > >     Our Ori-NF condition directly tests this ability. Prior works focused on oriented objects, but **they did not systematically examine whether the presence or absence of frontedness triggers different FoR selection strategies**—which is precisely the cognitive mechanism humans use. This capability is a necessary prerequisite for true object-centric reasoning and was not evaluated in prior work.
> > > > >
> > > > > 2. **Frames of Reference are a linguistic interface for human spatial communication**
> > > > >
> > > > >     FoR is not merely a computational choice—**it is a linguistic construct that reflects how humans express spatial understanding and resolve ambiguity**. To use FoR meaningfully in communication with humans, models must incorporate human strategies such as frontedness-driven FoR selection. Our evaluation examines whether models can adopt these human-centered strategies, enabling them to communicate spatial information in ways aligned with human reasoning rather than relying on surface-level patterns. Unlike prior benchmarks that only measure correctness, our work provides a diagnostic framework to analyze where and why this alignment breaks down, which is essential for human-AI interaction.

---

### Note · Authors · 2025-12-04

**Comment:**

We deeply appreciate you taking the time to review our manuscript and share your thoughtful feedback. Your observations have provided valuable guidance on improving the clarity, relevance, and impact of our work.

After reviewing. on your comments, we have decided to withdraw the paper to address your suggestions more thoroughly. This will allow us to refine our research and better highlight the significance of our work.

Thank you again for your time and for helping us strengthen our work.

**Withdrawal Confirmation:**

I have read and agree with the venue's withdrawal policy on behalf of myself and my co-authors.